# Distinct Urinary Metabolite Signatures Mirror In Vivo Oxidative Stress-Related Radiation Responses in Mice

**DOI:** 10.3390/antiox14010024

**Published:** 2024-12-27

**Authors:** Yaoxiang Li, Shivani Bansal, Baldev Singh, Meth M. Jayatilake, William Klotzbier, Marjan Boerma, Mi-Heon Lee, Jacob Hack, Keisuke S. Iwamoto, Dörthe Schaue, Amrita K. Cheema

**Affiliations:** 1Department of Oncology, Lombardi Comprehensive Cancer Centre, Georgetown University Medical Center, Washington, DC 20057, USA; yl814@georgetown.edu (Y.L.); sm3451@georgetown.edu (S.B.); bs1126@georgetown.edu (B.S.); mmj61@georgetown.edu (M.M.J.); 2Departments of Biochemistry, Molecular, and Cellular Biology, Georgetown University Medical Center, Washington, DC 20057, USA; wek11@georgetown.edu; 3Division of Radiation Health, Department of Pharmaceutical Sciences, University of Arkansas for Medical Sciences, Little Rock, AR 72205, USA; mboerma@uams.edu; 4Department of Radiation Oncology, David Geffen School of Medicine, University of California at Los Angeles, Los Angeles, CA 90024, USA; miheon1@gmail.com (M.-H.L.); jhack@mednet.ucla.edu (J.H.); kiwamoto@mednet.ucla.edu (K.S.I.); dschaue@mednet.ucla.edu (D.S.)

**Keywords:** radiation exposure, whole-body irradiation, partial-body irradiation, metabolomics, urine metabolomics, pathway analysis, machine learning, radiation effects, biomarkers

## Abstract

Exposure to ionizing radiation disrupts metabolic pathways and causes oxidative stress, which can lead to organ damage. In this study, urinary metabolites from mice exposed to high-dose and low-dose whole-body irradiation (WBI HDR, WBI LDR) or partial-body irradiation (PBI BM2.5) were analyzed using targeted and untargeted metabolomics approaches. Significant metabolic changes particularly in oxidative stress pathways were observed on Day 2 post-radiation. By Day 30, the WBI HDR group showed persistent metabolic dysregulation, while the WBI LDR and PBI BM2.5 groups were similar to control mice. Machine learning models identified metabolites that were predictive of the type of radiation exposure with high accuracy, highlighting their potential use as biomarkers for radiation damage and oxidative stress.

## 1. Introduction

Exposure to ionizing radiation, whether accidental, incidental, or through therapeutic modalities, can trigger a range of biological responses, including metabolic alterations [1,2,3,4,5]. In particular, ionizing radiation generates reactive oxygen species (ROS), which lead to oxidative stress—a critical mediator of cellular and metabolic dysfunction. This oxidative stress not only damages cellular macromolecules but also disrupts metabolic pathways, contributing to systemic responses post-radiation exposure [6]. Unlike the well-defined radiation damage to DNA/RNA [7,8,9], systemic alterations in metabolite levels and in metabolic pathways following radiation exposure in vivo are less understood, yet they have significant translational potential [10]. For instance, metabolites generated as the final products of cellular pathways in the body’s response to irradiation could act as potential biomarkers of radiation exposure, i.e., as biodosimeters [11]. Additionally, oxidative stress-related metabolic changes could provide insights into radiation-induced injuries and identify pathways that are amenable to antioxidant-based therapeutic interventions [12]. A detailed understanding of the biochemical and physiological mechanisms early in the aftermath of radiation exposure is also necessary for identifying individuals at risk of developing acute and late organ injuries. Ultimately, understanding radiation response and the metabolites that are associated with particular pathways may offer valuable insights into the mechanisms of radiation-induced injury and help design better mitigation strategies and novel therapeutic targets for individuals who might experience radiation toxicity [13,14].

In this study we used a hematopoietic acute radiation syndrome (H-ARS) model of adult C3Hf/HeJ/SED/KAM female mice in the context of different X-ray exposure scenarios to investigate urinary metabolic changes using untargeted and targeted metabolomics approaches. Our focus included identifying metabolic changes linked to oxidative stress pathways and exploring their relevance to radiation-induced injuries. We explored longitudinal alterations in metabolite levels and metabolic pathways for up to 90 days post exposure. Whole-body irradiation at high dose rate (WBI HDR) was compared to WBI at low dose rate (WBI LDR) and to partial-body irradiation with 2.5% bone marrow shielding (PBI BM2.5) in terms of metabolic signatures, with the overall goal to understand the broad as well as nuanced metabolic responses to irradiation. In particular, we sought to determine whether metabolic shifts associated with oxidative stress could explain dose-rate- and type-dependent radiation effects. We identified type- and dose-rate-dependent metabolic changes after radiation exposure and discovered late specific metabolites associated with organ damage. Using machine learning algorithms on the urine metabolomics data, we developed predictive models for the effects of radiation, enhancing our ability to pre-empt and possibly mitigate adverse outcomes. The overall methodology is depicted in Figure 1.

## 2. Material and Methods

### 2.1. Animal Procedures

Female, gnotobiotic C3Hf/HeJ/SED/KAM mice were bred in-house in our AAALAC-accredited animal facility and used for experiments at 4 months of age and 28.1 g ± 3.7 g body weight. All X-ray irradiation treatments were performed on anesthetized mice after i.p. injection of 80 mg/kg ketamine (KetaVed, ANADA#200–257, Vedco Inc., St. Joseph, MO, USA) and 4 mg/kg xylazine (Rompun, NADA#047-956; Dechra, Overland Park, KS, USA) (including sham-irradiated controls) using a Gulmay RS320 X-ray unit (Gulmay Medical Ltd., Camberley, Surrey, UK) operated at 300 kV and 10 mA with a permanent 4 mm Be filter plus 1.5 mm Cu and 3 mm Al beam filtration yielding a 3 mm Cu half-value layer (HVL) and a vertical beam with an FSD of 42.3 cm and a field size of 26.8 cm in diameter. X-rays were administered to unrestrained mice in the prone position at a dose rate of 1.312 Gy/min for whole-body irradiation at high dose rate (WBI HDR). An additional 1 mm Pb layer was placed at the beam outlet for reduced-dose-rate exposures (WBI LDR) at 0.0883 Gy/min and 2.9 mA. Partial-body irradiation with 2.5% bone marrow shielding (PBI BM2.5) was achieved with one leg including femur, tibia, and foot shielded under 1.4 cm Cerrobend blocks that yielded 99.5% attenuation.

The total dose for all three exposure scenarios was 6.791 Gy following LD30/30 estimates from prior prohibit analyses of H-ARS mortality, i.e., 30% mortality within 30 days. Dosimetry was performed with a Capintec ionization chamber calibrated to NIST standards, TLDs, and film (GAFCHROMIC EBT2, International Specialty Products, Wayne, NJ, USA) to confirm dose and field uniformity within 5% for WBI HDR and PBI BM2.5 and within 10% for WBI LDR. Animal health checks were performed once or twice daily during the first 30 days and then weekly until 500 days of age when heart, lung, kidneys, liver, and spleen were taken for histopathological examination.

Urine was collected onto sterile plastic sheets, transferred into microcentrifuge tubes and centrifuged at 12,000× *g* at 4 °C for 10 min. Supernatants were flash-frozen in LN and stored at −80 °C until shipment to Georgetown University on dry ice. MicroCT scanning (Siemens MicroCAT II CT Scanner, Knoxville, TN, USA) was used to evaluate lung damage based on loss in aerated volume. Mice were under continuous isoflurane (1–2% in O_2_) anesthesia while image acquisition was achieved over a total scan time of about 1 min plus 5–10 min image reconstruction. The 80 kVp 500 µA X-ray source was biased to 360 projections, an exposure time of 500 ms, detector bin mode of 4 × 4, and an effective pixel size of 0.2 mm. Regions of interest (ROIs) were drawn using the open-access Medical Image Data Examiner Amide (https://amide.sourceforge.net, accessed on 21 March 2022) to quantify the volume of aerated lung volume excluding the esophagus (3D-isocontour threshold < −400 HU). Blood was collected by cardiac puncture from euthanized animals into SST gold-top serum collection vials and after centrifugation was sent on dry ice for serum chemistry analysis (IDEXX BioAnalytics, Sacramento, CA, USA). Lungs were perfused with PBS followed by formalin, and together with heart, liver, kidney, and spleen tissues, were sent for processing and histopathological assessment by an expert pathologist who scored tissues on a 0–4 severity scale (IDEXX BioAnalytics, Columbia, MO, USA). Experimental protocols were approved by the Institutional Animal Care and Use Committee (IACUC) and premature humane euthanasia practices were strictly followed according to defined criteria and in compliance with NIH guidelines.

### 2.2. Chemicals

All LC-MS grade solvents (acetonitrile, methanol, isopropanol, and water) were purchased from Fisher Scientific (Pittsburgh, PA, USA). High-purity formic acid (99%), ammonium acetate, and LC–MS grade dichloromethane were obtained from Fisher Scientific. Debrisoquine and 4-nitrobenzoic acid were procured from Sigma-Aldrich (Saint Louis, MO, USA).

### 2.3. Targeted Metabolomics Analysis

The targeted metabolomics was performed in Multiple Reaction Monitoring (MRM) mode and quantitated 270 endogenous molecules using the QTRAP^®^ 5500 LC-MS/MS System (Sciex, MA, USA). Identification was based on MRM transitions and expected retention times, leveraging an established SCIEX method for metabolite detection. For this purpose, the sample preparation was kept similar to the preparation for untargeted analysis. A total of 5 μL of the prepared sample was injected into a Kinetex 2.6 μm 100 Å 100 × 2.1 mm column (Phenomenex, CA, USA) using a SIL-30 AC auto-sampler (Shimadzu, Kytoto, Japan) connected with a high-flow LC-30AD solvent delivery unit (Shimadzu, Kytoto, Japan) and a CBM-20A communication bus module (Shimadzu, Kytoto, Japan) online, with a QTRAP 5500 (Sciex, MA, USA) operating with polarity switching in both positive and negative modes. Binary solvents comprising water with 0.1% formic acid (solvent A) and acetonitrile with 0.1% formic acid (solvent B) were used. The extracted metabolites were resolved at 0.2 mL/min flow rate. The LC gradient conditions were as follows: initial—100% A, 0% B for 2.1 min; 14 min—5% A, 95% B until 15 min; 15.1 min—100% A, 0% B until 22.5 min. The auto-sampler and oven were kept at 15 °C and 30 °C, respectively. The source and gas settings for the method were as follows: curtain gas = 35; CAD gas = medium; ion spray voltage = 2500 V in positive mode and −4500 V in negative mode; temperature = 400 °C; nebulizing gas = 60; and heater gas = 70.

### 2.4. Untargeted Metabolomics Analysis

Untargeted metabolomics was performed using a UPLC-ESI-QTOF in TOFMS acquisition mode. For the untargeted LC-MS analysis, 20 μL urine was extracted with 80 μL 50:50 water/acetonitrile with internal standards (10 μL debrisoquine (1 mg/mL in ddH2O), 100 μL of 4-nitrobenzoic acid (1 mg/mL in methanol) per 10 mL). Samples were incubated at 4 °C for 20 min, then centrifuged at 15,493× *g* at 4 °C for 20 min. The supernatant was transferred to a MS vial and 1 μL of each was injected into a 130 Å, 1.7 μm, 2.1 mm × 50 mm Acquity UPLC BEH C18 column (Waters Corporation, MA, USA) maintained at 40 °C. The mobile phases consisted of water with 0.1% formic acid (solvent A) and acetonitrile with 0.1% formic acid (solvent B). The gradient was set as follows at a flow rate of 0.5 mL/minute: 0.0 min, 5% B; 0.5 min, 5% B; 4.0 min, 20% B; 8.0 min, 95% B; 9.0 min, 95% B; 9.1 min, 5% B; 11 min, 5% B. The column eluent was introduced directly into the G2 mass spectrometer by electrospray. Mass spectrometry was performed on a quadrupole–time-of-flight mass spectrometer operating in either negative or positive electrospray ionization. The positive mode had a capillary voltage of 3.00 kV and a sampling cone voltage of 30 V. The negative mode had a capillary voltage of 2.50 kV and had a sampling cone voltage of 30 V. The desolvation gas flow was set to 1000 L/hour and the desolvation temperature was set to 500 °C. The cone gas flow was 25 L/hour, and the source temperature was 120 °C. The data were acquired in the sensitivity MS mode with a scan time of 0.3 s. Accurate mass was maintained by infusing Leucine Enkephalin (556.2771 [M + H]^+^ and 554.2615 [M-H]^−^) via the Lockspray interface every 10 s. Data were acquired in centroid mode over a mass range of 50 to 1200 *m*/*z*.

### 2.5. Data Processing and Statistical Analysis

Both targeted and untargeted LC-MS-based metabolomics and lipidomics approaches were employed to quantify the abundance of urinary metabolites in both positive and negative ionization modes. Untargeted data were preprocessed using an in-house implementation of XCMS [15] (Scripps Institute, CA, USA) to generate a peak intensity table. Metabolites were annotated using the METLIN database, with matches confirmed by accurate mass measurements. Metabolite intensities were normalized to internal standards and QC samples, and further processed using data pre-processing techniques including log transformation and Pareto scaling. The differential expression of each metabolite was calculated using a rigorous statistical analysis approach, incorporating an unpaired *t*-test with a significance threshold of *p* < 0.05. To ensure the robustness and accuracy of the data, stringent quality control measures were implemented, including the application of a 20% coefficient of variation filter criteria [16] and missing value imputation techniques such as the half-min algorithm. Additionally, analytical drift was corrected using the quality control sample-based robust LOESS (locally estimated scatterplot smoothing) signal correction (QC-RLSC) [17] method, resulting in highly reliable and accurate data for the identification of biomarkers associated with radiation subgroups. Additionally, the R package MOFA2 (version 1.16.0) [18] was used to conduct the multi-omics factor analysis.

### 2.6. Predictive Modeling

Predictive modeling was performed using the R package caret (version 7.0) [19] to classify radiation exposure using urine metabolomics data. The dataset was split into training and testing sets in a 50:50 ratio to ensure balanced representation. Feature selection was performed using the Boruta algorithm [20]. Neural Network (N-Net) and XG-Boost algorithms were applied for classification. Optimal hyperparameters were determined using leave-one-out cross-validation (LOOCV). Model performance was evaluated using accuracy, precision, recall, confusion matrices, and AUC-ROC values. Representative boxplots of metabolite intensities were plotted to illustrate variability across exposure groups. The AUC curves comparison and optimal cut-off point phase involved plotting the AUC curves of the different models in one figure for comparison. The optimal cut-off point for classification was identified, and the predicted probability index plot was generated.

## 3. Results

### 3.1. Metabolomic Analyses Yield a Robust Radiation Signature

Adult, female C3Hf/HeJ/SED/KAM mice were randomly assigned to the following four treatment groups representing different radiation exposure scenarios: WBI HDR (*n* = 15), WBI LDR (*n* = 15), PBI BM2.5 (*n* = 14), or sham-irradiated control mice (*n* = 20). Urine samples were collected over a period of 90 days, beginning with baseline data collection 14 days prior to treatment, followed by Day 2, 30, and 90 post-irradiation to capture immediate and delayed metabolic responses to radiation exposure. A combination of LC-MS-based targeted and untargeted approaches were used to understand the differential radiation responses. Given the known role of oxidative stress in radiation biology, we also explored metabolic pathways associated with ROS; as such, the in-depth understanding of molecular changes will enable the development of medical countermeasures with antioxidant properties to reverse metabolic shifts and thus alleviate radiation-induced acute and late organ damage.

The matched comparison of radiation versus sham exposures over time exhibited robust radiation effects with most metabolites showing downregulation on Day 2, as illustrated in the volcano plots (Figure 2A). The early downregulation of metabolites on Day 2 likely reflects oxidative damage and disruptions in pathways related to redox balance and antioxidant defense repair mechanisms. As detailed in Appendix A, by Day 30, the number and magnitude of changes in metabolite abundance had drastically reduced in the WBI LDR and PBI BM2.5 groups. By Day 90, all the irradiated groups presented metabolite signatures that more closely matched with those of the sham-irradiated controls (Appendix A). In fact, these trends were also reflected in the principal component analysis (PCA) comparing the urinary metabolomic profiles of sham-treated vs. irradiated mice (Figure 2B,C). A clear group separation was confirmed on Day 2 post exposure but less so by Day 30, when the PBI BM2.5 and WBI LDR groups significantly overlapped with controls while the WBI HDR mice did not. These differences may be partially attributable to persistent oxidative stress and impaired repair mechanisms in the WBI HDR group, leading to prolonged metabolic disturbances. By Day 90 post-irradiation, PCA indicated minimal divergence in metabolite levels among all groups, suggesting that the majority of the radiation-induced metabolic alterations had been resolved and/or a new metabolic equilibrium had been established (Appendix A). The resolution of early oxidative stress-associated metabolic changes by Day 90 supports the hypothesis that redox homeostasis is gradually restored post-irradiation. The initial (acute) separation highlights the immediate metabolic alterations triggered by radiation across all groups, but different repair or recovery kinetics for different types of exposure scenarios also emerged, with WBI HDR driving particularly prolonged metabolic changes, i.e., presumably more persistent damage.

### 3.2. Pathway Analysis Identifies Radiation-Induced Pathway Perturbations

We performed pathway analysis using the Mummichog Python package (version 2.06) [21] to identify metabolic pathways perturbed by radiation exposure (Figure 3). Radiation exposure resulted in the upregulation of antioxidant pathways including retinol metabolism in the PBI BM2.5 group, sialic acid metabolism in the WBI LDR group, and vitamin B5 metabolism in the WBI HDR group, possibly as a response to combat radiation-induced oxidative stress. These antioxidant pathways likely reflect the activation of protective mechanisms aimed at mitigating ROS-mediated damage caused by radiation. On the other hand, multiple pathways associated with energy, redox, and amino acid metabolism were downregulated in the three groups. The downregulation of redox-related pathways may indicate an early depletion of repair mechanisms necessary for maintaining oxidative balance post-radiation exposure.

We also observed the disruption of tyrosine metabolism across all radiation exposure groups (Figure 3A–C). This could have wide-ranging implications in vivo, affecting several essential biochemical pathways that rely on tyrosine, including the synthesis of thyroid hormones [22], melanin, and several key neurotransmitters such as L-DOPA, dopamine, adrenaline, and noradrenaline, in line with physiological and cognitive changes [23,24]. Another intriguing pathway that ranks high in all radiation groups is galactose metabolism; D-galactose is known to promote mitochondrial function and cellular tolerance to oxidative stress [25]. This suggests that galactose metabolism may play a compensatory role in cellular defense against oxidative damage induced by radiation.

Taken together, early metabolic changes in response to radiation may provide insights into a coordinated systemic response to radiation and biochemical aberrations that can be targeted to protect or reverse radiation damage and/or assist recovery. The identification of pathways related to oxidative stress and antioxidant defenses underscores the critical role of redox balance in shaping metabolic responses to radiation.

### 3.3. Multi-Modal Evaluation of Radiation-Induced Organ Damage

Next, we used multi-omics factor analysis (MOFA) to identify key factors that capture the multi-dimensional variance within complex datasets, such as metabolite concentrations, lipid levels, and other measured outcomes of radiation exposure including serum chemistry and CT aerated volume, all measured at the same time points as the urine metabolites. Initially, the dataset was centered along a one-dimensional axis at zero that revealed a balanced distribution of factors with minimal skewness or bias, suggesting the amenability of this dataset for MOFA (Figure 4A).

Using MOFA and a significance threshold of *p* < 0.05, Factors 2, 3, 5, and 7 were identified as significantly correlated with radiation tissue damage (Figure 4B). The X-axis represents measured endpoints and the Y-axis illustrates the degree of influence or correlation each sample or observation holds with a specific latent factor. For example, Factor 1 was strongly linked to spleen and kidney degeneration, while Factor 5 was associated with kidney inflammation as well as premature death, highlighting their close correlation with different but specific types of organ damage observed post-irradiation. Interestingly, the prediction of radiation type was associated with multiple Factors including 2, 3, 5 and 7, which corroborates the notion that radiation exposure causes a pleiotropic response systemically. This systemic response likely includes oxidative stress, which contributes to organ-specific damage by disrupting metabolic and cellular homeostasis.

Next, we examined a subset of factors to enable a biological understanding of radiation outcomes. For example, Figure 4C highlights key urinary metabolomic markers associated with Factor 5 that show a strong correlation with radiation-induced kidney inflammation, including mannose, alpha-D-glucose, NAD, glucosamine-6-phosphate, and butyrylcarnitine. Several of these metabolites are linked to oxidative stress pathways, suggesting their role in mediating tissue-specific damage induced by radiation. Additionally, it includes CT aerated volume and relevant serum chemistry endpoints such as conjugated bilirubin and creatinine. Figure 4D, focusing on Factor 7, delineates other significant markers in the context of radiation subgroups. Taken together, MOFA indicates that radiation-induced organ damage may depend on a combination of several orthogonal factors that when combined provide a more specific prediction model.

Further analysis of MOFA2 Factors 5 and 7 revealed sub-structures in our multi-modal dataset that are best illustrated in inter-factor plots with top-weighted features, especially when combining Factors 5 and 7 (Figure 5). These plots visually represent the relationships between factors and their influence on data variability. Key features linked to oxidative stress and antioxidant defenses, such as cis-aconitate and alpha-tocopherol, were identified among the top-weighted features, underscoring the importance of redox balance in shaping radiation responses. The top-weighted features in the MOFA2 Factor 5 and 7 analyses show the significance of these factors and highlight those metabolic alterations that are most affected by radiation. Specifically, in this dataset, they reveal unique patterns for each of the radiation subgroups and suggest that the combined effects of Factors 5 and 7 are superior at distinguishing these groups than each factor alone. Overall, the analysis reveals distinct metabolic changes following radiation exposure in vivo, including significant alterations in pathways related to oxidative stress, inflammation, and cellular metabolism. Increased levels of alpha-tocopherol, 5-methionine, SAH, and cis-aconitate have been associated with organ-specific damage in the kidney, linked to premature aging and inflammation across tissues and indicative of persistent oxidative stress that contributes to prolonged metabolic alterations and tissue-specific damage.

Next, group classification was confirmed through the correlation analysis of individual MOFA2 factor features. The top 12 weighted metabolomic features in mouse urine, associated with Factor 7, each demonstrated significant correlations with this factor, as reflected in their respective *p*-values (Figure 6). Both positive and negative correlations were noted, indicating an increase or a decrease, respectively, in metabolite levels with respect to Factor 7, underscoring the nuanced nature of metabolic alterations induced by radiation. Importantly, MOFA2 correlation analysis revealed distinct patterns for each treatment group, supporting the idea that Factor 7 can differentiate radiation exposure types. In Figure 6, each sub-panel represents a scatter plot of a specific metabolite’s normalized intensity (Y-axis) against the values of Factor 7 (X-axis). Positive and negative correlations indicate whether the metabolite levels increase or decrease with respect to Factor 7. For example, metabolites like phenyllactate and glucosamine-6-phosphate show strong positive correlations with Factor 7, while others like methyladenosine exhibit negative correlations. The presence of metabolites linked to redox balance, such as glucosamine-6-phosphate, further supports the role of oxidative stress in shaping group-specific responses to radiation. These unique patterns among various metabolites highlight how different treatment groups respond to radiation exposure, showing specific metabolic profile changes that are dependent on the type and dose of radiation.

### 3.4. Predictive Modeling of Radiation Exposure

One of the goals of the study was to leverage the metabolomics data for the classification of radiation exposure. We used a multi-model-based machine learning approach to stratify controls from irradiated animals in different study groups including ’WBI LDR’, ’WBI HDR’, and ’PBI BM2.5’ at the Day 2 time point. To ensure balanced representation, the data were split evenly into training and testing sets. We used algorithms, including Neural Network (N-net) and Gradient Boosting (XG-boost), for classification to obviate bias. Feature selection was performed using Boruta (version 8.0.0), and each model was trained to differentiate between individual radiation exposure groups and the control.

The predictive models incorporated a 6-metabolite panel for Day 2 and 19-metabolite panel for Day 30, including the metabolites atrolactate, cAMP, hydroxyphenyllactate, and succinylcarnitine, among others. Many of these metabolites are involved in oxidative stress-related pathways, such as succinylcarnitine, which plays a role in mitochondrial function, and indolepropionate, a known antioxidant with protective effects against radiation-induced damage. Details of the complete panel, including *p*-values and fold changes, are provided in Appendix A. To illustrate the variability in metabolite intensities across exposure groups, representative box plots in Appendix A highlight the differences in metabolic profiles among the WBI-LDR, WBI-HDR, and PBI-BM2.5 groups. For instance, metabolites such as carbamoylaspartate and phenyllactate showed pronounced upregulation in the WBI-HDR group, suggesting heightened oxidative stress and disrupted metabolic homeostasis under high-dose-rate conditions. As illustrated in Figure 7, the area under the receiver operating characteristic (AUROC) scores and performance metrics (e.g., Sensitivity, Specificity, Precision, Recall, F1 Score, and Balanced Accuracy) showed a high accuracy of exposure classification. For Day 2, XG-boost generally outperformed N-net for the PBI-BM2.5 (AUROC 0.961) and WBI-HDR (AUROC 0.956) groups. By Day 30, the N-net model showed improvement across classes, while XG-boost also showed a slight increase in AUROC values for all classes. These performance metrics underscore that each model has distinct strengths in predicting specific radiation types, with both models exhibiting balanced accuracy close to or above 0.7 across classes. However, the difficulty in accurately classifying WBI-LDR may reflect subtler metabolic changes related to oxidative stress, as indicated by lower levels of redox-active metabolites such as succinylcarnitine in this group compared to WBI-HDR.

## 4. Discussion

The present study provides a comprehensive view of metabolic signatures in urine collected from mice after different types of radiation exposures and their impact on organ damage. We used a mouse model of exposure to different radiation types in conjunction with a LC-MS-based comprehensive urinary metabolomics profiling approach to identify significant changes in metabolite levels and metabolic pathways following exposure. The underlying idea was to identify early indicators of organ damage as well as identify changes within the first 48 h of irradiation to develop models of exposure classification.

Our analysis revealed that the most pronounced metabolic changes occurred at Day 2 post-radiation, underscoring the immediate impact of radiation exposure on the body’s metabolic processes. These early changes likely reflect oxidative stress, a key mechanism underlying radiation-induced damage, which disrupts redox balance and alters metabolic homeostasis. WBI HDR was found to induce sustained metabolic dysregulation up to Day 30 post-radiation, while both LBI HDR and PBI BM2.5 demonstrated fewer significant changes in metabolic profiles at this time point and beyond. This finding highlights the role of dose rate in modulating oxidative stress, as high-dose-rate exposures may exacerbate ROS production and the associated metabolic disruptions. These findings suggest that the dose rate and the hematopoietic reserve matter in terms of the extent and duration of the metabolic radiation signal. Urine metabolite profiles showed unique temporal changes in WBI vs. PBI, particularly for metabolites such as methionine sulfoxide and norvaline. The assumption is that some, if not all, of these systemic metabolic changes originate from specific organ dysfunction. Persistent alterations in oxidative stress-related metabolites provide further evidence of systemic redox imbalance contributing to organ dysfunction.

The identification of key radiation-induced metabolites, such as methionine sulfoxide, succinylcarnitine, and carbamoylaspartate, underscores their role in oxidative stress and organ-specific damage. These metabolites provide potential biomarkers for radiation exposure and injury. Importantly, the dysregulation of metabolites like succinylcarnitine suggest disrupted mitochondrial function, while the dysregulation of methionine sulfoxide reflected oxidative protein damage. Taken together, these findings imply that antioxidant-based therapies targeting oxidative stress pathways could mitigate radiation-induced injuries and improve recovery. Further validation in preclinical and clinical settings is essential to establish these metabolites as robust biomarkers for early detection and therapeutic intervention. Several of these metabolites are directly involved in oxidative stress pathways, suggesting their potential role in mitigating or exacerbating organ injury. These metabolites may play a role in organ injury. Furthermore, the alterations in multiple key metabolic pathways we identified highlight the complex and multifaceted nature of radiation-induced injury and emphasize the need for further validation to correlate with biological effects.

More research is needed, particularly on the role of important metabolites such as carbamoylaspartate, methionine sulfoxide, and succinylcarnitine. These metabolites, known for their roles in redox regulation and mitochondrial function, are particularly relevant for understanding the oxidative stress response following radiation exposure. Further validation will be required to confirm a link between these metabolic alterations and their pathophysiological relevance that may point to new therapeutic options and/or their use as accurate biomarkers for organ-specific defects. Another limitation of this study is that only female mice were used; as such, it is imperative to test the generalizability of these findings using male cohorts.

One of the novel approaches described herein pertains to the use of MOFA for the prediction of organ damage; we found that factors comprising multimodal data were better predictors of organ damage while metabolite markers were very effective markers of radiation exposure in the acute phase. MOFA analysis also highlighted oxidative stress-related pathways as critical contributors to radiation-induced tissue injury, particularly through Factors 5 and 7. The use of multiple machine learning algorithms, including Boruta, N-Net, and XG-boost on urine metabolomics data has showcased their potential value in predicting radiation types (WBI/PBI). These models successfully predicted radiation effects and their subgroups, introducing a novel method for assessing radiation-induced metabolic changes. The multi-modal evaluation of radiation-induced organ damage, illustrated in Figure 5, highlighted significant correlations between various factors, particularly the top features of Factors 5 and 7 in radiation-induced kidney inflammation subgroups. Key features linked to oxidative stress, such as succinylcarnitine and glucosamine-6-phosphate, were prominent contributors to these factors, underscoring their relevance in predicting tissue-specific damage. These features, comprising urinary metabolomics markers, CT aerated volume, and serum chemistry markers, emphasize the importance of a multi-modal approach in assessing radiation-induced organ damage, providing a holistic view of systemic changes—metabolic or otherwise—caused by radiation. Although the focus of this study was to evaluate metabolic dysregulation in a biodefense scenario, some of the pathways identified in this study, including retinol metabolism, vitamin B5 metabolism, and sialic acid metabolism, associated with cellular defense mechanisms against oxidative stress are suggestive of radiation response irrespective of radiation type. For example, studies focused on understanding the molecular mechanism underlying FLASH-radiation therapy (defined as the ultrafast delivery of radiation to achieve better tumor control while sparing normal tissue toxicity) suggest reduction in oxidative stress, redox balance and mitochondrial dysfunction in normal tissues [26,27]. These data suggest that the upregulation of antioxidant pathways observed in the WBI HDR and WBI LDR groups in this study could parallel the mechanisms underlying the protective effects of FLASH irradiation. Further investigation into how these pathways are modulated under ultra-high-dose-rate conditions is warranted.

## 5. Conclusions

This study has provided a comprehensive exploration of the systemic metabolic alterations induced by different types and doses of radiation. Utilizing both targeted and untargeted metabolomics approaches, we have identified significant changes in metabolite levels and metabolic pathways post-radiation exposure. Many of these changes were associated with oxidative stress and its downstream effects, reflecting a critical role for redox imbalance in shaping the metabolic response to radiation, thus underscoring the potential of antioxidants for use as radiation protectors and/or mitigators. The temporal pattern of these alterations, with the most pronounced changes observed at Day 2 post-radiation, underscores the immediate and lasting impact of radiation on the body’s metabolic processes.

## Figures and Tables

**Figure 1 antioxidants-14-00024-f001:**
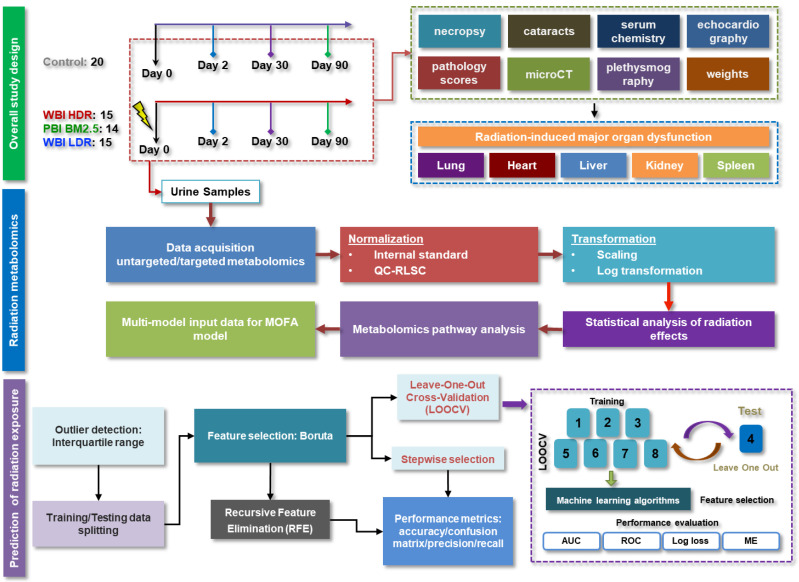
An overview of the study design, depicting the experimental workflow. Mice were subjected to different radiation exposures (WBI HDR, WBI LDR, PBI BM2.5), followed by urine collection at multiple time points (Days 0, 2, 30, and 90). The schematic outlines sample collection, metabolomics analysis, pathway identification, and predictive modeling to determine oxidative stress-related metabolic changes and identify biomarkers of radiation damage.

**Figure 2 antioxidants-14-00024-f002:**
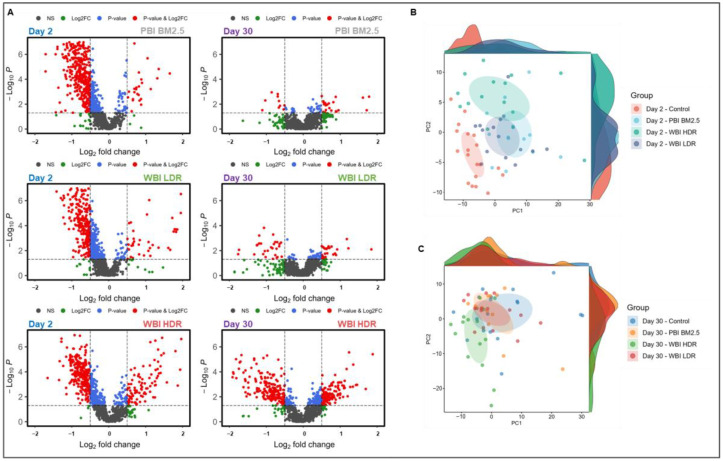
Radiation exposure induces robust and dose-dependent urinary metabolite alterations. Untargeted metabolomics analysis reveals significant metabolite alterations following radiation exposure. (**A**): Volcano plots indicate downregulation of most metabolites on Day 2. (**B**,**C**): Principal component analysis (PCA) demonstrates group separation on Day 2, with WBI HDR showing greatest separation from controls. By Day 30, WBI LDR and PBI BM2.5 groups exhibit metabolic recovery, while WBI HDR remains distinct. At Day 90, metabolic profiles converge with controls across all groups, indicating resolution of radiation-induced perturbations.

**Figure 3 antioxidants-14-00024-f003:**
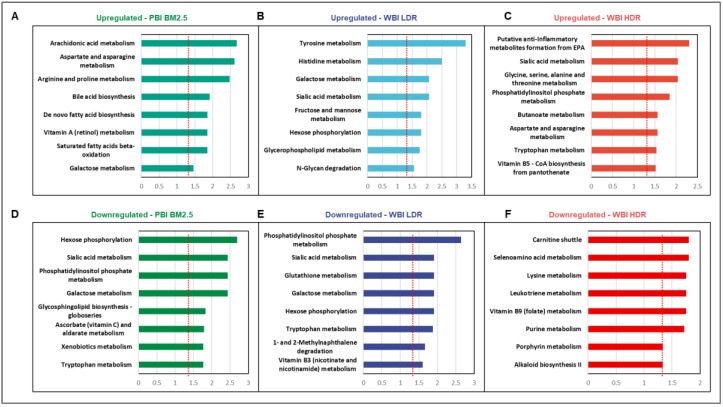
Pathway analysis identified radiation-specific metabolic perturbations, including oxidative stress-related pathways. Metabolic pathway analysis revealed radiation-specific changes across exposure groups. (**A–C**): Upregulated pathways included retinol metabolism (PBI BM2.5), sialic acid metabolism (WBI LDR), and vitamin B5 metabolism (WBI HDR), indicative of antioxidant responses to oxidative stress. (**D–F**): Downregulated pathways encompassed energy metabolism, redox balance, and amino acid metabolism in WBI HDR group, reflecting sustained metabolic dysregulation.

**Figure 4 antioxidants-14-00024-f004:**
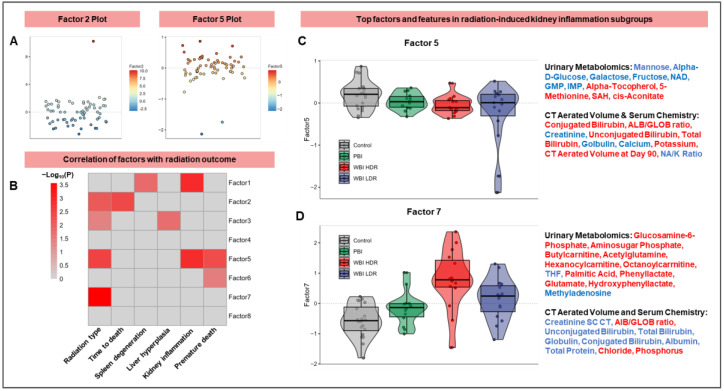
MOFA identifies factors linked to radiation-induced organ damage and oxidative stress. MOFA highlights key factors driving metabolic and phenotypic variance post-radiation. (**A**): Scatter plots confirm minimal skewness across factors, supporting robustness of dataset. (**B**): Correlation heatmap identifies significant associations between Factors 2, 3, 5, and 7 and radiation-induced outcomes, such as kidney inflammation and spleen degeneration. (**C**,**D**): Violin plots show distribution of Factors 5 and 7, highlighting metabolites positively (red) and negatively (blue) correlated with radiation effects. Oxidative stress-related metabolites including cis-aconitate, glucosamine-6-phosphate are enriched in these factors.

**Figure 5 antioxidants-14-00024-f005:**
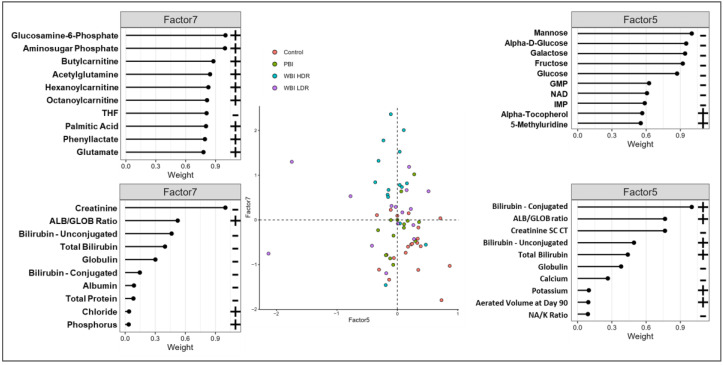
Inter-factor analysis reveals oxidative stress-associated metabolic features. Inter-factor plots between MOFA Factors 5 and 7 display relationships among key metabolic features. Top-weighted features include cis-aconitate, alpha-tocopherol, and glucosamine-6-phosphate, indicative of oxidative stress and antioxidant responses. These features distinguish radiation exposure types and provide insights into persistent metabolic alterations.

**Figure 6 antioxidants-14-00024-f006:**
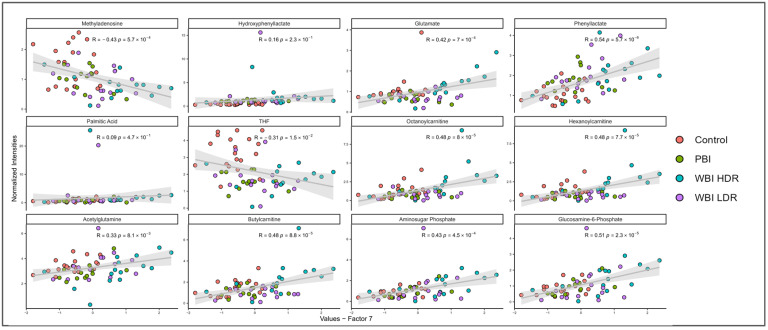
Correlations of top metabolites with MOFA Factor 7 highlight radiation-specific metabolic responses. Scatter plots show correlations between Factor 7 and normalized intensities of key metabolites. Points are color-coded by treatment group (Control, PBI BM2.5, WBI HDR, and WBI LDR). Positive correlations including phenyllactate and glucosamine-6-phosphate and negative correlations including methyladenosine reflect group-specific metabolic shifts linked to oxidative stress. These metabolites serve as indicators of radiation exposure and oxidative imbalance.

**Figure 7 antioxidants-14-00024-f007:**
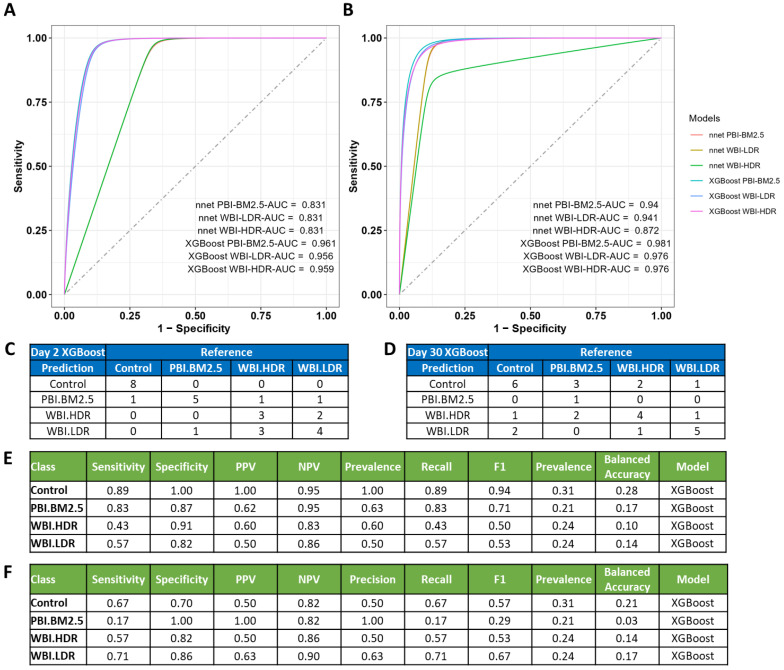
Machine learning models predict radiation exposure types based on urinary metabolite profiles. Performance of machine learning models (N-net and XGBoost) in classifying radiation exposure types. (**A**,**B**): Area under the receiver operating characteristic curves demonstrate high predictive accuracy, especially for PBI BM2.5 and WBI HDR groups at Days 2 and 30. (**C**,**D**): Confusion matrix summarizes classification outcomes, showing robust performance of XGBoost. (**E**,**F**): Model performance metrics (Sensitivity, Specificity, Precision, Recall, F1 Score, Balanced Accuracy) showing XGBoost’s high predictive capability.

## Data Availability

All data generated or analyzed during this study are included in this published article (and its Appendix A).

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
