# Peer review of "Distinct Urinary Metabolite Signatures Mirror In Vivo Oxidative Stress-Related Radiation Responses in Mice"

_antioxidants, 2024, doi:10.3390/antiox14010024_

Round 1
Reviewer 1 Report
The manuscript entitled “Distinct Urinary Metabolite Signatures Mirror in vivo Oxidative Stress Related Radiation Responses in Mice” (antioxidants-3346425) by Yaoxiang et al. is well written and interesting paper. In this work, authors used metabolomics platform including targeted and untargeted approaches to identify metabolic changes following radiation exposure and discovered late specific metabolites associated with organ damage in a mouse model. Using MOFA analysis, a novel approach, the authors confirmed that oxidative stress-related pathways are key pathways contributing to radiation-induced tissue damage. The obtained results showed that the most pronounced metabolic changes occurred on the second day after radiation and suggest that oxidative stress is the key mechanism responsible for radiation-induced damage. In my opinion, the presented results will be of interest to many researchers working on radiation response and altered metabolic pathways, providing deeper insight into the mechanisms of radiation-induced damage. Therefore, I believe that the paper is valuable and will undoubtedly interest the journal’s readers.
However, there are some points that should be improved:
The use of numerous abbreviations throughout the text requires searching for their meaning. This makes reading difficult and makes the work difficult to read. Although the full meaning of the abbreviation is given on first use, I strongly recommend grouping all the abbreviations together with their full meaning in words in a separate section. This would be beneficial for the readers
2In the methods section:
- 1) The paragraph 2.4. Detailed information on used deuterated lipid internal standards is missing. Furthermore, I am confused because the purpose of this study was metabolomic analysis of urine samples, and some parts of the methods section refer to plasma samples (paragraph 2.4., lines 156-158). It should be revised.
- 2) A representative total ion chromatogram (TIC) for the 21 lipid classes should be provided (even in supplementary), showing the retention time ranges at which each lipid class was eluted.
- 3) On what basis was the identification of individual lipid species and other metabolites made?
- 4) While chromatographic conditions are very well described in the MS part some crucial information are not included. Although ion source condition are provided the mode in which MS was operated should be given for both targeted and nontargeted approach
- 5) DDA analysis was used for metabolite and lipid profiling? It should be also mentioned in the method section.
Author Response
Comments 1: The manuscript entitled “Distinct Urinary Metabolite Signatures Mirror in vivo Oxidative Stress Related Radiation Responses in Mice” (antioxidants-3346425) by Yaoxiang et al. is well written and interesting paper. In this work, authors used metabolomics platform including targeted and untargeted approaches to identify metabolic changes following radiation exposure and discovered late specific metabolites associated with organ damage in a mouse model. Using MOFA analysis, a novel approach, the authors confirmed that oxidative stress-related pathways are key pathways contributing to radiation-induced tissue damage. The obtained results showed that the most pronounced metabolic changes occurred on the second day after radiation and suggest that oxidative stress is the key mechanism responsible for radiation-induced damage. In my opinion, the presented results will be of interest to many researchers working on radiation response and altered metabolic pathways, providing deeper insight into the mechanisms of radiation-induced damage. Therefore, I believe that the paper is valuable and will undoubtedly interest the journal’s readers.
Response 1: We thank the reviewer for their positive and encouraging feedback regarding the quality and impact of our work. We are pleased that the results were found to be valuable and of interest to researchers investigating radiation responses and metabolic pathways. We greatly appreciate the reviewer’s comments highlighting the clarity and relevance of our study.
Comments 2: However, there are some points that should be improved:
The use of numerous abbreviations throughout the text requires searching for their meaning. This makes reading difficult and makes the work difficult to read. Although the full meaning of the abbreviation is given on first use, I strongly recommend grouping all the abbreviations together with their full meaning in words in a separate section. This would be beneficial for the readers.
Response 2: Thank you for this helpful suggestion. We have created a new "Abbreviations" section at the end of the manuscript that includes all abbreviations and their full meanings. This change ensures easier readability and reference for readers.
Comments 3: In the methods section:
1) The paragraph 2.4. Detailed information on used deuterated lipid internal standards is missing. Furthermore, I am confused because the purpose of this study was metabolomic analysis of urine samples, and some parts of the methods section refer to plasma samples (paragraph 2.4., lines 156-158). It should be revised.
2) A representative total ion chromatogram (TIC) for the 21 lipid classes should be provided (even in supplementary), showing the retention time ranges at which, each lipid class was eluted.
3) On what basis was the identification of individual lipid species and other metabolites made?
4) While chromatographic conditions are very well described in the MS part some crucial information are not included. Although ion source condition are provided the mode in which MS was operated should be given for both targeted and nontargeted approach
5) DDA analysis was used for metabolite and lipid profiling? It should be also mentioned in the method section.
Response 3:
Response 3.1: Thank you for pointing this out. Upon review, we have removed Section 2.4, as targeted lipidomics (TL) data were not included in this study. The earlier version of the manuscript inadvertently referenced TL methodology, which was initially considered but not part of the final analysis. The study focuses on targeted metabolomics and untargeted metabolomics data from urine samples, as described in Sections 2.3 and 2.5. We have ensured that all methods described align with the data presented in the study.
Response 3.2: Thank you for the suggestion. Since targeted lipidomics data are not part of this study, a TIC specific for the 21 lipid classes are not applicable.
Response 3.3: Thank you for pointing this out. We have revised the manuscript to clarify the identification process for targeted and untargeted approaches. For targeted metabolomics, identification was based on MRM transitions and expected retention times, using an established SCIEX method for metabolite detection. For untargeted metabolomics, metabolites were annotated using the METLIN database.
Response 3.4: We appreciate this observation. The MS operation mode and source parameters have now been specified for both targeted and untargeted metabolomics. Targeted analysis was performed in MRM mode, while untargeted analysis used TOFMS mode. Updated in Section 2.3: "Targeted metabolomics was performed with polarity switch in both positive and negative mode" Updated in new Section 2.4: " Untargeted metabolomics was performed using a UPLC-ESI-QTOF in TOFMS acquisition mode. For the untargeted LC-MS analysis, 20 μL urine was extracted with 80 μL 50:50 water/acetonitrile with internal standards" and "Mass spectrometry was performed on a quadrupole-time-of-flight mass spectrometer operating in either negative or positive electrospray ionization. Positive mode had a capillary voltage of 3.00 kV and a sampling cone voltage of 30 V. Negative mode had a capillary voltage of 2.50 kV and had a sampling cone voltage of 30 V. The desolvation gas flow was set to 1000 L/hour and the desolvation temperature was set to 500°C. The cone gas flow was 25 L/hour and the source temperature was 120 °C. The data was acquired in the sensitivity MS mode with a scan time of 0.3 seconds. Accurate mass was maintained by infusing Leucine Enkephalin (556.2771 [M+H] + and 554.2615 [M-H]-) via the Lockspray interface every 10 seconds. Data was acquired in Centroid mode over a mass range of 50 to 1200 m/z. "
Response 3.5: Thank you for this observation. We have explicitly mentioned that TOFMS analysis was used for untargeted profiling Methods Section 2.4. (Same as Response 3.4)
Reviewer 2 Report
The abstract contains complex jargon and dense phrasing. Simplify the language in the abstract to make key findings accessible , avoiding overly technical jargon.
The methodology includes a mix of technical details and general descriptions, making it inconsistent. Ensure a uniform level of detail across all experimental methods
The discussion does not delve deeply into the implications of the findings for therapeutic applications or biomarker development
Figure 1 lacks a clear explanation of its role and insights it provides. Provide clear explanations and context for all figures and tables, ensuring they support the narrative effectively.
Provide clear explanations and context for all figures and tables, ensuring they support the narrative effectively.
Author Response
Reviewer 2
Comments 1: The abstract contains complex jargon and dense phrasing. Simplify the language in the abstract to make key findings accessible, avoiding overly technical jargon.
Response 1: Thank you for pointing this out. We agree with this comment and have simplified the abstract to make the key findings more accessible and clearer while avoiding overly technical terms.
Manuscript Update:
- Page 1, Abstract, Entire abstract rewritten for clarity, accessibility, and reduced technical jargon.
Comments 2: The methodology includes a mix of technical details and general descriptions, making it inconsistent. Ensure a uniform level of detail across all experimental methods
Response 2: We have updated the entire materials and methods section and ensured a consistent level of detail across all subsections.
Revised Sections: Section 2.1, Section 2.2, Section 2.3, Section 2.4, Section 2.5, Section 2.7
Comments 3: The discussion does not delve deeply into the implications of the findings for therapeutic applications or biomarker development
Response 3: Thank you for this suggestion. We have expanded the discussion section to emphasize the translational relevance of our findings for therapeutic interventions and biomarker development.
Revised Section:
- Page 12, Section 4: The identification of key radiation-induced metabolites, such as methionine sulfoxide, succinylcarnitine, and carbamoylaspartate, underscores their role in oxidative stress and organ-specific damage. These metabolites provide potential biomarkers for radiation exposure and injury. Importantly, metabolites like succinylcarnitine highlight disrupted mitochondrial function, while methionine sulfoxide reflects oxidative protein damage. Such findings suggest that antioxidant-based therapies targeting oxidative stress path-ways could mitigate radiation-induced injuries and improve recovery. Further validation in preclinical and clinical settings is essential to establish these metabolites as robust biomarkers for early detection and therapeutic intervention.
Comments 4: Figure 1 lacks a clear explanation of its role and insights it provides. Provide clear explanations and context for all figures and tables, ensuring they support the narrative effectively.
Response 4: We appreciate this feedback and have added clearer explanations for Figure 1 to provide a clear explanation of its role in the study and how it supports the narrative.
Revised Sections:
- Page 2, revised legend: Figure 1: Overview of the study design, depicting the experimental workflow. Mice were subjected to different radiation exposures (WBI HDR, WBI LDR, PBI BM2.5), followed by urine collection at multiple time points (Days 0, 2, 30, and 90). The schematic outlines sample collection, metabolomics analysis, pathway identification, and predictive modeling to determine oxidative stress-related metabolic changes and identify biomarkers of radiation damage.
Comments 5: Provide clear explanations and context for all figures and tables, ensuring they support the narrative effectively.
Response 5: We have carefully revised the legends for all figures and tables to ensure they provide sufficient context and support the narrative effectively
Revised legends:
Figure 2. Radiation exposure induces robust and dose-dependent urinary metabolite alterations.
Untargeted metabolomics analysis revealed significant metabolite alterations following radiation exposure. Panel A: Volcano plots indicate downregulation of most metabolites on Day 2. Panels B and C: Principal component analysis (PCA) demonstrates group separation on Day 2, with WBI HDR showing the greatest separation from controls. By Day 30, WBI LDR and PBI BM2.5 groups exhibit metabolic recovery, while WBI HDR remains distinct. At Day 90, metabolic profiles converge with controls across all groups, indicating resolution of radiation-induced perturbations.
Figure 3. Pathway analysis identifies radiation-specific metabolic perturbations, including oxidative stress-related pathways. Metabolic pathway analysis revealed radiation-specific changes across exposure groups. Panels A–C: Upregulated pathways include retinol metabolism (PBI BM2.5), sialic acid metabolism (WBI LDR), and vitamin B5 metabolism (WBI HDR), indicative of antioxidant responses to oxidative stress. Panels D–F: Downregulated pathways encompass energy metabolism, redox balance, and amino acid metabolism in the WBI HDR group, reflecting sustained metabolic dysregulation.
Figure 4. MOFA identifies factors linked to radiation-induced organ damage and oxidative stress. MOFA highlights key factors driving metabolic and phenotypic variance post-radiation. Panel A: Scatter plots confirm minimal skewness across factors, supporting the robustness of the dataset. Panel B: Correlation heatmap identifies significant associations between Factors 2, 3, 5, and 7 and radiation-induced outcomes, such as kidney inflammation and spleen degeneration. Panels C and D: Violin plots show the distribution of Factors 5 and 7, highlighting metabolites positively (red) and negatively (blue) correlated with radiation effects. Oxidative stress-related metabolites including cis-aconitate, glucosamine-6-phosphate are enriched in these factors.
Figure 5. Inter-factor analysis reveals oxidative stress-associated metabolic features. Inter-factor plots between MOFA Factors 5 and 7 display relationships among key metabolic features. The top-weighted features include cis-aconitate, alpha-tocopherol, and glucosamine-6-phosphate, indicative of oxidative stress and antioxidant responses. These features distinguish radiation exposure types and provide insights into persistent metabolic alterations.
Figure 6. Correlations of top metabolites with MOFA Factor 7 highlight radiation-specific metabolic responses. Scatter plots shows correlations between Factor 7 and the normalized intensities of key metabolites. Points are color-coded by treatment group (Control, PBI BM2.5, WBI HDR, and WBI LDR). Positive correlations including phenyllactate, glucosamine-6-phosphate and negative correlations including methyladenosine reflect group-specific metabolic shifts linked to oxidative stress. These metabolites serve as indicators of radiation exposure and oxidative imbalance.
Figure 7. Machine learning models predict radiation exposure types based on urinary metabolite profiles. Performance of machine learning models (N-net and XGBoost) in classifying radiation exposure types. Panels A and B: Area under the receiver operating characteristic curves demonstrate high predictive accuracy, especially for PBI BM2.5 and WBI HDR groups at Days 2 and 30. Panels C and D: Confusion matrix summarize classification outcomes, showing robust performance of XGBoost. Panels E and F: Model performance metrics (Sensitivity, Specificity, Precision, Recall, F1 Score, Balanced Accuracy) showing XGBoost’s high predictive capability.